# Drug Metabolite Cluster-Based Data-Mining Method for Comprehensive Metabolism Study of 5-hydroxy-6,7,3′,4′-tetramethoxyflavone in Rats

**DOI:** 10.3390/molecules24183278

**Published:** 2019-09-09

**Authors:** Yuqi Wang, Xiaodan Mei, Zihan Liu, Jie Li, Xiaoxin Zhang, Shuang Lang, Long Dai, Jiayu Zhang

**Affiliations:** 1School of Chinese Pharmacy, Beijing University of Chinese Medicine, Beijing 100029, China (Y.W.) (X.M.) (Z.L.) (J.L.) (X.Z.) (S.L.); 2School of Pharmacy, BIN ZHOU Medical University, Yantai 260040, China

**Keywords:** 5-hydroxy-6,7,3′,4′-tetramethoxyflavone (HTF), metabolites, drug metabolite clusters (DMCs), ultra-high performance liquid chromatography coupled with high-resolution mass spectrometry, data-mining methods

## Abstract

The screening of drug metabolites in biological matrixes and structural characterization based on product ion spectra is among the most important, but also the most challenging due to the significant interferences from endogenous species. Traditionally, metabolite detection is accomplished primarily on the basis of predicted molecular masses or fragmentation patterns of prototype drug metabolites using ultra-high performance liquid chromatography coupled with high-resolution mass spectrometry (UHPLC-HRMS). Although classical techniques are well-suited for achieving the partial characterization of prototype drug metabolites, there is a pressing need for a strategy to enable comprehensive drug metabolism depiction. Therefore, we present drug metabolite clusters (DMCs), different from, but complementary to, traditional approaches for mining the information regarding drugs and their metabolites on the basis of raw, processed, or identified tandem mass spectrometry (MS/MS) data. In this paper, we describe a DMC-based data-mining method for the metabolite identification of 5-hydroxy-6,7,3′,4′-tetramethoxyflavone (HTF), a typical hydroxylated-polymethoxyflavonoid (OH-PMF), which addressed the challenge of creating a thorough metabolic profile. Consequently, eight primary metabolism clusters, sixteen secondary metabolism clusters, and five tertiary metabolism clusters were proposed and 106 metabolites (19 potential metabolites included) were detected and identified positively and tentatively. These metabolites were presumed to generate through oxidation (mono-oxidation, di-oxidation), methylation, demethylation, methoxylation, glucuronidation, sulfation, ring cleavage, and their composite reactions. In conclusion, our study expounded drug metabolites in rats and provided a reference for further research on therapeutic material basis and the mechanism of drugs.

## 1. Introduction

The science of drug metabolism has developed significantly over the past two decades. Its central role in today’s field of medicine has matured to the point where drug metabolism could be applied to elucidate the action mechanisms and create a holistic picture of drugs. In recent years, numerous studies have been conducted in the drug metabolism and metabolism-related fields [1]. It shows that prototype drugs might undergo bio-transformation in vivo and produce metabolites with various bioactivities that possess diverse metabolic pathways [2,3]. Therefore, metabolite identification has already been used to aid diagnoses, discover bioactive constituents, and study the integrative mechanism of drugs [4,5].

Ideally, different technologies could be combined to both identify metabolites and create a whole picture of drug phenotypes. However, metabolite identification has been regarded as one of the key challenges in current mass spectrometry (MS)-based untargeted metabolism studies. Moreover, there is still no comprehensive and appropriate database or platform to mine, distinguish, and analyze drug metabolites. While some natural product databases can provide a useful glimpse into metabolite dereplication, this is usually limited to the identification of known prototype drugs. Experiments generating data that are much more proximal to a drug profile such as the Dictionary of Natural Products [6], Metlin [7], and Massbank [8], this can be too expensive or limit data analyses to several individual spectra, and a challenge remains regarding distinguishing the causal origin of metabolites for researchers. Meanwhile, the bottleneck occurring in metabolite identification is to trace both targeted and untargeted drug metabolites as the MS signals generating from those of interest are likely to be overwhelmed in the significant interferences from the background or endogenous matrix given their low concentrations. Thus, there is a pressing need for a novel analytical strategy, platform, or database to perform MS-based metabolite identification.

The previously reported methods for monolithic metabolite profiling have already been expanded dramatically and have evolved through various stages to the point where certain applications have reached the market as diagnostic kits [9,10,11]. However, as with any evolving method, there are a number of issues mainly focusing on the metabolic reactions that have the closest relationship with prototype drugs. In fact, drug metabolites are not only produced directly from the prototype drug, but are also yielded by many intermediate metabolites that undergo a variety of sequential biotransformation processes. That is to say, previous studies, particularly those that have concentrated on the direct metabolites of prototype drugs, will definitely result in incomprehensive metabolite determination.

Consequently, we proposed a novel methodology named drug metabolite clusters (DMCs), which was employed to mine the information about drugs and their metabolites on the basis of raw, processed, or identified tandem mass spectrometry (MS/MS) data. The prototype drug was reacted through a specialized drug metabolizing enzymatic system to produce primary metabolites (intermediate metabolites) according to certain metabolic regulations or preferences [12], and then accordingly formed primary metabolite cluster centers and primary clusters of DMC. The primary metabolites were further metabolized in light of the corresponding reactions to generate secondary metabolite cluster centers and secondary metabolite clusters. By analogy, the new rank of metabolism cluster centers and metabolism clusters can emerge continuously, as long as the intermediate metabolites can reach the appropriate concentration range under the biotransformation circumstances. This means that each identified metabolite can become a new metabolite cluster center, and then corresponding reactions occur to produce new metabolites/cluster centers. Regardless of which cluster center is exposed, the other metabolite information associated with the cluster center will be unlocked, making it easy to capture all the relevant metabolite information. Since successive drug metabolic reactions extend around the centers gradually, all the cluster centers are related according to the appropriate relationships, and then eventually form the DMC (shown in Figure 1).

In this study, 5-hydroxy-6,7,3′,4′-tetramethoxyflavone (HTF), attributed to hydroxylated- (OH-PMFs), was adopted as a study case to present the proposed DMC data-mining method for comprehensive metabolite characterization and metabolic pathway clarification. Meanwhile, a novel strategy based on the DMC method coupled with multiple post-acquisition, data-processing techniques, and structural elucidation approaches for the profiling and identification of metabolites, particularly those metabolized from the intermediate metabolites by using ultra-high performance liquid chromatography coupled with high-resolution mass spectrometry (UHPLC-HRMS), was established. As demonstrated by the results, a total of 106 HTF metabolites (HTF included) with different structures were positively identified or tentatively assigned. Concurrently, the DMC of HTF in in vivo metabolic pathways were established on the basis of those detected metabolites.

## 2. Results

In this study, we explored a novel and integrated strategy based on the DMC-based data mining method combined with multiple numerous parallel technologies, which was characterized by data-acquisition approaches, data-processing techniques, and a structural elucidation approach (shown in Figure 2). Our established strategy has been utilized for screening and identifying HTF metabolites, particularly microconstituents, on a hybrid LTQ-Orbitrap mass spectrometer.

First, high resolution extracted ion chromatogram (HREIC) and multiple mass defect filtering (MMDF) technologies were adopted to preliminarily screen both the targeted and untargeted candidates after the acquisition of high quality accurate raw mass data in the positive as well as negative ion modes. Simultaneously, the prototype drug was selected as the object to predict metabolites, according to the multiple metabolic pathways template, and the primary cluster centers were established. Second, DMC was developed for the complete screen and prediction of HTF metabolites. Third, the full scan-parent ions list-dynamic exclusion (FS-PIL-DE) data acquisition method was applied to ensure the overall data integrity and quality by obtaining a specific ESI-MS^n^ datasets. Then, a combination of data mining methods including neutral loss fragments (NLFs) and diagnostic product ions (DPIs) was performed to rapidly confirm and identify the HTF metabolites. Finally, the metabolic pathways of HTF were proposed, grounded on the DMC and bibliography.

### 2.1. Establishment of HREIC and MMDF Based Data Processing Method

In the present study, the combination of MMDF and HREIC technologies was used for online data acquisition to trace all of the primary metabolites, known as primary cluster centers. HREIC can accurately and controllably detect the constituents with predictive metabolite weights. Hence, HR-ESI-MS^1^ analysis was performed on the HRMS instrument with a resolution of 30,000 and the full-scan MS datasets were processed by setting the mass error of the predictive molecular weights within ± 5 ppm.

Furthermore, three MDF templates were set in parallel as follows: (1) the prototype drug type filter template; (2) glucuronide conjugation type template; and (3) sulfate conjugation type template. The prototype drug type filter template was based on the composition of HTF (C_19_H_18_O_7_) with a mass defect shift of 113.0778 mDa. The other two types of conjugate filters, glucuronic acid and sulfonic acid, were also established including HTF + glucuronide with a mass defect shift of 137.3406 mDa and HTF + sulfate with a mass defect shift of 62.0672 mDa. Each MDF template window was set to ±50 mDa around the mass defect of an applied filter template over a mass range of ±50 Da around the mass of filter template.

### 2.2. Establishment of DMC

In the process of metabolism, prototype drugs usually generate the intermediate metabolites first, then the intermediate metabolites undergo further metabolic reactions. For instance, the prototype drug C_19_H_18_O_7_ was selected as the screening object by using the conventional metabolite templates established according to the literature reports and the pre-experiment results. Additionally, a series of metabolites were selected. Second, based on the primary metabolites, eight primary cluster centers were defined. Third, every possible center was also chosen as an object to screen according to the conventional metabolite templates. Depending on the information obtained above, the structures of drug metabolites (i.e., new cluster centers) were elucidated and identified positively or tentatively. After the application of the above methods to the obtained data, the majority of interference ions would be remarkably reduced. Furthermore, the HTF metabolite ions of interest could be clarified from the entire MS datasets, which meant that it was easier to trace all of the potential metabolites than the unprocessed spectra.

### 2.3. Combined Post-Acquisition Data Mining Methods

HTF, a compound based on the backbone of 2-phenyl-1-benzopyran-4-one, consists of four methoxy groups and one hydroxy group. Therefore, Retro-Diels-Alder (RDA) rearrangement and the neutral loss of methyl and methoxy groups should be its major fragmentation pathway in CID-MS/MS experiment. To obtain the elimination patterns, we found the basic laws from the experimental exploration by analyzing, contrasting, and concluding the mass fragmentation behaviors of nine different PMF reference standards (shown in Table 1). Due to the differences among the core structures of OH-PMFs, these characteristic RDA rearrangements are much more easily formed to enhance the stability of the carbon ions.

In the ESI-MS^2^ spectra, all nine PMF reference standards yielded the characteristic product ions due to the neutral loss of one or more CH_3_· groups. Thus, we utilized mass inclusion/exclusion criteria of n × 15 for the entire mass range of datasets to efficiently screen metabolites from gigantic MS/MS datasets and then screen out the metabolite ions of interest. At the same time, a series of DPIs, representing a certain parent nucleus or substitution groups, were used as the Appendix A to determine or tentatively assign the corresponding chemical families from these candidates, which considerably enhanced the identification of unusual and minor metabolites.

By means of the previous reports and preliminary analyses, HTF metabolites could be categorized into three major groups on the basis of the possible scenarios of their core structures (shown in Figure 3): (I) polymethoxyflavone-structure-based metabolites, the DPI at *m*/*z* 151 + X (X = molecular weight of substituent groups, such as 14, 16, 30, for CH_2_, O, OCH_2_, etc)., yielded by RDA rearrangement occurred on positions 0, 4-position of C-ring; (II) polymethoxyflavanone-structure-based metabolites, the DPI at *m*/*z* 211 yielded by RDA rearrangement occurred on positions 1, 3-position of C-ring with methyl group transfer from B ring to A ring making carbon ions more stable form; and (III) polymethoxychalcone-structure-based metabolites, the DPI at *m*/*z* 221 yielded by the RDA rearrangement occurred on positions 1, 3-position of C-ring, and another DPI at *m*/*z* 197 yielded by RDA rearrangement occurred on positions 1, 4-position of C-ring, making the carbon ions a more stable form.

#### 2.3.1. Structural Assignment of the Representative Polymethoxyflavones

Metabolites **M0**, **M9**, and **M10,** which possess the same protonated [M + H]^+^ ion at *m*/*z* 359.11256 (C_19_H_19_O_7_, error within ±5 ppm) or deprotonated [M − H]^−^ ion at *m*/*z* 357.09692 (C_19_H_17_O_7_, error within ±5 ppm), were eluted at 8.86, 8.37, and 12.83 min, respectively. By comparing the full-scan ESI-MS^n^ spectra and retention time with the obtained reference standards, M0 was unequivocally deduced to be the prototype drug. In positive ion mode, the subsequent product ions were observed with the successive losses of CH_3_·, yielding typical product ions at *m*/*z* 344 and *m*/*z* 329, respectively. Meanwhile, the fragment ion at *m*/*z* 298 originating from the ion at *m*/*z* 359 was interpreted as neutral, eliminating CH_3_·, CO, and the simultaneous loss of H_2_O via the neighboring hydrogen transferring to C-5 hydroxy group of HTF. Moreover, *m*/*z* 298 generated the product ion at *m*/*z* 283 because of the successive loss of CH_3_·. Furthermore, the DPI at *m*/*z* 151 (C_8_H_7_O_3_) yielded by RDA rearrangement occurred on 0, 4-position of C-ring, further confirmed our conjecture. The accurate mass weight and major product ions of **M9** and **M10** coincided with those of **M0**, indicating that they could be deduced as HTF isomers (shown in Appendix A).

Metabolites **M5**–**M7**, with protonated [M + H]^+^ ion at *m*/*z* 345.09687 (C_18_H_17_O_7_, error within ±5 ppm) or deprotonated [M − H]^−^ ion at *m*/*z* 343.08117 (C_18_H_15_O_7_, error within ± 5 ppm), were individually eluted at 10.77, 10.97, and 11.26 min. They were 14 Da less than HTF in either the positive or negative ion modes, indicating they might be demethylated products of HTF. The deprotonated ion yielded the DPIs at *m*/*z* 328 and *m*/*z* 313 by the successive elimination of CH_3_·. Then, the fragment ion at *m*/*z* 312 was formed after subsequently losing OCH_3_· from the ion at *m*/*z* 343, while the fragment ion at *m*/*z* 300 was yielded after losing CO from the ion at *m*/*z* 328 (shown in Appendix A). Meanwhile, the minor ion at *m*/*z* 151 (C_8_H_7_O_3_) was also observed due to RDA fragmentation from the 0, 4-position of the C-ring, which could be adopted to further confirm our above-mentioned deduction. Therefore, we determined that **M5**–**M7** were determined as isomeric dihydroxyl-triamethoxyflavones.

#### 2.3.2. Structural Assignment of the Representative Polymethoxyflavanones

Metabolites **M14** and **M15** with protonated [M + H]^+^ ion at *m*/*z* 361.12817 (C_19_H_21_O_7_, error within ±5 ppm) or deprotonated [M − H]^−^ ion at *m*/*z* 359.11362 (C_19_H_19_O_7_, error within ±5 ppm), were in turn eluted at 7.33 min and 12.23 min, respectively. These were 2 Da more than HTF, whether in the positive or negative ion modes, which further generated the prominent product ion at *m*/*z* 211 (C_10_H_11_O_5_) as a base peak ion in their MS^2^ spectra in positive ion mode. It could be deduced that the dominating fragmentation pathway was RDA cleavage from the 1, 3-position of the C-ring. Meanwhile, the minor ion at *m*/*z* 151 (C_8_H_7_O_3_) was also detected, due to the RDA fragmentation from the 0, 4-position of the C-ring. A series of DPIs at *m*/*z* 196 and *m*/*z* 183 were also detected in their MS^2^ and MS^3^ spectra, given the respective loss of CH_3_· and CO (shown in Appendix A). Therefore, the particular pathways and DPIs could be adopted as a shortcut to rapidly distinguish polymethoxyflavanones from general flavones. Hence, **M14** and **M15** could be attributed to be monohydroxyl-tetramethoxyflavanones.

#### 2.3.3. Structural Assignment of the Representative Polymethoxychalcones

**M34**, **M35**, **M36**, and **M37** possessing the experimental protonated [M + H]^+^ ion at *m*/*z* 377.12312 (C_19_H_21_O_8_, error within ±5 ppm) and deprotonated [M − H]^−^ ion at *m*/*z* 375.10747 (C_19_H_19_O_8_, error within ±5 ppm) were detected at 10.65, 10.98, 11.14, and 17.02 min, respectively. Their dissociation pathways of [M + H]^+^ ion were similar to each other on the whole. The RDA cleavage at bond B of its [M + H]^+^ ion to yield the DPI at *m*/*z* 221 (C_12_H_13_O_4_) and at bond A to yield the minor ion at *m*/*z* 197 (C_9_H_9_O_5_) could also be detected in their positive MS^2^ spectra. Meanwhile, the fragment ions detected from the loss of 15 (CH_3_·), 18 (H_2_O), 28 (CO), and 30 (2CH_3_·) could be also adopted as DPIs for polymethoxylated chalcones in their MS^2^ and MS^3^ spectra (shown in Appendix A). Thus, **M34**, **M35**, **M36**, and **M37** were subsequently identified as isomeric dihydroxyl-tetramethoxychalcones.

#### 2.3.4. Structural Assignment of the Conjugate-Metabolites

Sulfate conjugation is one of the most commonly seen conjugation reactions. **M72**, which eluted at 8.54 min, showed the [M − H]^−^ ion at *m*/*z* 453.04857 (C_19_H_17_O_11_S, −0.063 ppm), 80 Da more than HTF. The DPI at *m*/*z* 373 [M – H − SO_3_]^−^ was observed in its MS/MS spectrum. Meanwhile, the other fragment ions at *m*/*z* 358 [M – H − SO_3_ − CH_3_·]^−^ and *m*/*z* 343 [M − H − SO_3_ − 2CH_3_·]^−^ were also detected, which indicates the occurrence of the sulfation reaction. Therefore, this was putatively identified as the sulfated product of HTF.

Glucuronide conjugation is another important type of conjugating reaction. **M78** exhibited the deprotonated ion at *m*/*z* 533.12897 (C_25_H_25_O_13_, −0.35 ppm), which was 176 Da more than that of HTF. Its ESI-MS^2^ spectrum gave the base peak ion at *m*/*z* 357 via the loss of the glucuronide moiety (176 Da) from *m*/*z* 533, which indicated the presence of the glucuronide group in its structure. Moreover, the ion at *m*/*z* 357 yielded DPIs at *m*/*z* 342 and *m*/*z* 327 in the ESI-MS^3^ spectrum due to the neutral loss of CH_3_· and 2CH_3_·, respectively. Therefore, **M78** was supposed as a glucuronide group conjugate to the HTF. The details of proposed metabolites are shown in Appendix A.

#### 2.3.5. Summary of HTF Metablolites

As a result, a total of 106 predictive metabolites (19 potential metabolites included) were distinguished in rat urine and plasma samples after oral administration of HTF based on DMC. Finally, 87 metabolites were accurately identified. The MS^n^ spectra of 19 potential metabolites were not detected, which might be due to the mass signals being embedded among those of irrelevant endogenous ions, or a low dosage of oral administration prototype drug. The correlative data are summarized in Table 2 and Table 3, and the HREIC spectra of the HTF metabolites are illustrated in Figure 4.

In light of the results, the DMC of HTF was eventually formed and included eight primary cluster centers, sixteen secondary primary cluster centers, and five tertiary primary cluster centers. Furthermore, the main metabolic reaction types included oxidation (mono-oxidation and di-oxidation), methylation, demethylation, methoxylation, glucuronidation, sulfation, ring cleavage, and their composite reactions in in vivo biotransformation (shown in Figure 5). In addition, it should be noted that some cracked ring and rearrangement products were produced such as flavanones and chalcones. Among them, 52 metabolites were found in urine and 80 metabolites in plasma (shown in Appendix A). After using the DMC-based data-mining method, more metabolites were added, among which 67 metabolites were attributed to OH-PMFs including 15 polymethoxylated flavanones and 12 polymethoxylated chalcones (shown in Figure 6).

## 3. Discussion

HTF is one of the major OH-PMFs, existing mainly in the citrus genus such as *Citrus sinensis*, *Citrus mangshanesis*, and *Citrus clementina*, which displayed significant antioxidant activities and showed remarkable inhibition effects on the B16 cell lines at a concentration range from 6.25 to 50 μg/mL [13]. OH-PMFs, the flavonoid subclass in which all or almost hydroxyls are capped by methylation, have drawn more and more attention recently because accumulating evidence has demonstrated that OH-PMFs have many stronger health-promoting biological activities and higher oral bioavailability than their permethoxylated counterparts [14]. OH-PMFs have been proven to exhibit a broad spectrum of biological properties including anti-carcinogenic [15,16], anti-coagulation [17], anti-inflammatory [18,19], and anti-oxidant activities [20,21]. Many studies have also shown that OH-PMFs could induce the bacteria in the lower digestive tract to produce corresponding enzymes, which hydrolyze glycosides into aglycones to facilitate their absorption, so as to strengthen the effect for diseases [22].

Notwithstanding, up to now, no related literature reports are available that describe the in vivo metabolism of HTF. In this study, we extended the application of the DMC method based on UHPLC-HRMS to study the formation of HTF metabolites. Our goal will be to synthesize the identified metabolites as the protocol in quantitatively measuring the distribution of HTF and its metabolites for a better understanding of toxicity and efficacy.

Recently, the high mass accuracy and rapidly decreasing costs of UHPLC-HRMS such as orbital trapping analyzers are enabling their wide application for metabolite identification [23,24]. It offers high chromatographic resolution with the exact mass measurement for the MS data, and then provides the acquisition of full-scan MS spectra and product-ion spectral datasets for both targeted and untargeted compounds with significant advantages concerning sensitivity, fidelity, and speed. A single MS experiment can collect thousands of MS/MS spectra in minutes, however, for a given prototype drug, not all may be of interest [25]. Thus, the major difficulty in drug metabolism is usually the extraction of relevant information from the immensely complex sample profiles recorded. Recognizing the need for achieving data mining, increasing the metabolism coverage, and identifying metabolites rapidly, we presented the DMC method, a network methodology throughout the drug metabolism.

DMC is a correlation and visualization approach that can detect sets of compounds from related metabolites (cluster center), even when the compounds themselves are not metabolized directly. With the application of DMC, most of the noise signals in the spectra are removed and the list of metabolites is enriched. In the process of metabolism, the prototype drug usually generates the intermediate metabolites first, which means that candidate molecule identities of the DMC primary cluster centers are formed. Once the metabolite (cluster center) has been ascertained, the new template can be accordingly established. Certain metabolic reactions performed around each center present more and more new centers, so some uncommon drug metabolites could be much more comprehensively detected. Thus, the metabolism clusters spring up over and over again in an endless spiral, with the DMC becoming more correct, more comprehensive, and richer each time. Eventually, it is possible to achieve metabolite prediction of the whole metabolic process.

It was well-known that the metabolites in the same metabolism cluster usually share a similar core substructure. Hence, setting the metabolism cluster center is the key step of the DMC-based method. Here, any expected primary cluster centers are screened using an improved template (shown in Appendix A) derived from a wide range of biotransformation pathways. Examples of these reactions and associated references were adapted from a previously published literature review and empirical research, thereby enhancing applications to predict the performance of plasma metabolite profiling.

HREIC and MMDF are the most effective data mining methods to ensure the centers because they can readily predict metabolite molecular weights and elemental compositions derived from the accurate mass measurements and interpret accurate MS/MS spectra in an entire LC/MS dataset. HREIC, which provides qualitative metabolite profiles by extracting the data for a particular molecular or fragment ion, is often utilized for simple bioreaction products and predictable metabolites such as methylation, hydroxylation, sulfate conjugation, etc. [26]. Nevertheless, unusual metabolites and metabolites with low abundance that are not readily predicted present more of a challenge.

Therefore, the simple implementation of MMDF was employed to remove false positives generated from HREIC in the detection of metabolites in complicated bioreaction products. The MMDF data dependent acquisition not only use prototype drugs as a filter template, but also apply more than one multiple mass defect windows selected mass ranges based on the mass defects of the prototype drug ion and its observed cluster centers as well as possible metabolic reactions to acquire MS/MS spectra for various types of drug metabolites [27].

Due to the significant differences in content and occasionally poor chromatographic separation, metabolites with low concentrations from the full-scan mass chromatograms cannot be detected and their product-ion MS/MS acquisitions cannot be triggered in most cases, especially when excess quantities of endogenous components are co-eluted. Therefore, in order to obtain the extensive fragmentation behavior of target constituents, particularly microconstituents, and ensure comprehensive metabolite detection, FS-PIL-DE and data-dependent acquisition methods were applied to obtain specific ESI-MS^n^ datasets based on DMC, which could target all of the predictable constituents that have the same molecular weight, regardless of the site of substitution positions and fragmentations [28]. Meanwhile, the above approach based on DMC also failed to obtain partial components (potential metabolites) from MS^n^ data primarily because of the distinct characteristics of the metabolite ions.

Access to the complete MS datasets (from HRMS^1^ to MS^n^) of metabolic candidates is a prerequisite in this quest for metabolite structural characterization. However, metabolite identification is still the challenge in drug metabolism profiling. Therefore, a combination of various data mining tools including NLFs and DPIs were chosen to identify HTF metabolites [29,30]. In general, one compound often splits into two parts in CID mode, and thus produces product ions and neutral fragments, which are complementary in structural elucidation. Particularly, in DMCs, metabolites originating from the prototype drug through specific metabolic pathways were characterized by the inheritance and presented a similar skeleton with different positional substitution groups, so it is possible to undergo corresponding fragmentation pathways in CID mode as well as produce specific DPIs with regular NLFs. A series of DPIs represent a certain parent nucleus and NLFs stand for substitution groups or typical groups, which can be used as the characteristic peaks to select the corresponding metabolites.

As a result, the structures of 106 HTF metabolites were elucidated based on the DMC data mining methodology including 87 identified metabolites and 19 potential metabolites. According to the phenomenon of isomerism, we found that the transference of hydroxy and methoxy groups were more easily formed due to distinctive structure of HTF with multiple hydroxy and methoxy groups. For example, although there were five oxidation replacement sites, seven oxidation products of HTF were detected. Among these different isomers, the positions of the hydroxy group and methoxy group should vary slightly. Additionally, the methodology of DMC is a more logical and hierarchical identification technique, which could guide metabolite screening, enrich metabolic theories, and reveal metabolic processes reasonably.

Cluster centers of the HTF DMC were perceived as the main essential ingredients in the metabolism study such as mono-hydroxy-tetra-methoxyflavanone, mono-hydroxy-tetra-methoxychalcone, mono-hydroxy-penta-methoxyflavone, and so on. With the prolongation of metabolic reactions, implying the generation of metabolite clusters, metabolite clusters have been increasingly developed and related, and are not completely independent. Conversely, they have links, influence, and even intersection among each other as well as a radial metabolism net, which confirmed the hypotheses of DMC put forth in this thesis.

When Archimedes said, “Give me a place to stand and I will move the Earth,” he was referring to leverage. Here, we said “Give me a prototype drug to metabolize, and I will unlock its DMC”. Unlike previous studies, DMC provides a metabolite-led net where drug metabolites can be detected, analyzed, and identified rapidly and accurately. Specifically, it is available to simultaneously enable the qualitative analysis of multiple metabolites to be performed on the prototype drug and metabolism cluster centers with different metabolism pathways in the same analysis. In addition, it is a potentially powerful methodology to significantly enhance comprehensive metabolite profiling and provide an opportunity to alter the process of metabolite identification dramatically. Last but not least, DMC enables the discovery of drug metabolites, the characterization of biosynthetic pathways, the understanding of the chemistry of ecological interactions, and the development of metabolism bioinformatics methods in complex systems such as chemical medicine, traditional Chinese medicines, veterinary drugs, and so on.

## 4. Materials and Methods

### 4.1. Chemicals and Materials

Seven PMFs (polymethoxyflavonoids) reference standards (P-1, P-2, P-4, P-5, P-6, P-8, P-9) were previously extracted, isolated, and identified from *Murraya paniculata* (L.) in our laboratory. Their structures were fully elucidated through a comparison of their spectral data (ESI-MS and ^1^H, ^13^C NMR) with published literature values. Their purities were all determined to be higher than 98% by HPLC-UV. 5,7,3′,4′,5′-pentamethoxyflavanone (P-3) was purchased from the National Institutes for Food and Drug Control (Beijing, China), and 5-demethylnobiletin (P-7) was purchased from Chengdu Biopurify Phytochemicals Ltd. (Chengdu, China). All structures for these nine PMF reference standards are shown in Table 4 and Figure 7.

HPLC grade acetonitrile, methanol, and formic acid (FA) were purchased from Thermo Fisher Scientific (Fair Lawn, NJ, USA). All other chemicals of analytical grade were available at the work station, Beijing Chemical Works (Beijing, China). Deionized water used throughout the experiment was purified by a Milli-Q Gradient Å 10 System (Millipore, Billerica, MA, USA). Grace PureTM SPE C18-low solid-phase extraction cartridges (200 mg/3 mL, 59 µm, 70 Å) for the pretreatment of biological samples were supplied by Grace Davison Discovery Science^TM^ (Deerfield, IL, USA).

### 4.2. Animals and Drug Administration

Eight Sprague-Dawley (SD) rats (male, 200 ± 20 g) were purchased from Beijing Weitong Lihua Experimental Animals Company (Beijing, China) and kept under controlled environmental conditions (temperature: 24 ± 2 °C; relative humidity: 70 ± 5%; day and night alternation time: 12 h light/dark cycle) with free food intake and water consumption. After one week of acclimatization, rats were randomly divided into two groups: the drug group (*n* = 4) and the control group (*n* = 4). Rats in the drug group were given a dose of 250 mg HTF, which was suspended in 0.5% carboxymethyl cellulose sodium (CMC–Na) solution per kilogram of body weight, by oral gavage at. Equivalent volumes of the 0.5% CMC–Na aqueous solution was administrated to rats in the control group. The animal protocols were approved by the institutional Animal Care and Use Committee at the Beijing University of Chinese Medicine. The animal facilities and protocols complied with the Guide for the Care and Use of Laboratory Animals (USA National Research Council, 1996).

### 4.3. Sample Collection and Preparation

Before the experiment, all rats were fasted for 12 h with free access to water prior to the experiment. Blood samples (0.5 mL) were taken from the suborbital venous plexus of rats at 0.5, 1, 2, and 4 h post-administration. Each sample was centrifuged (3500 rpm, 4 °C) for 10 min to separate the plasma samples. Finally, all of the homogeneous biological samples from the same group were merged into a collective sample. The blank plasma sample was prepared from rats in the control group by using the same method as that used in the drug group. All of the above plasma samples were stored at −80 °C before subsequent analysis.

An approach using a SPE cartridge for the precipitation and concentration of protein and solid residue was performed to prepare the biological samples. The SPE cartridges were pre-activated with methanol (5 mL) and deionized water (5 mL), respectively. Subsequently, plasma samples (1 mL) were added to the SPE cartridges. Finally, the SPE cartridges were successively washed with deionized water (5 mL) and methanol (3 mL). The methanol eluate was concentrated and evaporated with nitrogen at room temperature. The residue was redissolved in 100 µL 10% acetonitrile solution and centrifuged for 15 min (13,500 rpm, 4 °C). All supernatants were used for further instrumental analysis.

### 4.4. Instrument and Conditions

A DIONEX Ultimate 3000 UHPLC system (Thermo Fisher Scientific, MA, USA) was connected to a hybrid LTQ-Orbitrap HRMS (Thermo Scientific, Bremen, Germany) equipped with an electrospray ionization source. The chromatographic separation was carried out at 40 °C using a Waters ACQUITY BEH C18 column (2.1 × 100 mm i.d., 1.7 μm; Waters Corporation, Milford, MA, USA). The mobile phase consisted of 0.1% FA aqueous solution (A) and acetonitrile (B) at a flow rate of 0.2 mL/min, and the linear gradient procedure was described as follows: 0–2 min, 5–20% B; 2–27 min, 20–85% B; 27–30 min, 85% B.

HRMS and MS/MS spectra were obtained using LTQ-Orbitrap MS with the optimized operating parameters set as follows. Positive ion mode: sheath gas (nitrogen) flow rate of 40 arb, auxiliary gas (nitrogen) flow rate of 20 arb, capillary temperature of 350 °C, spray voltage of 4.0 kV, capillary voltage of 25 V, and tube lens voltage of 110 V. Negative ion mode: sheath gas (nitrogen) flow rate of 40 arb, auxiliary gas (nitrogen) flow rate of 20 arb, capillary temperature of 350 °C, spray voltage of 3.0 kV, capillary voltage of −35 V, and tube lens voltage of −110 V. The metabolites were detected by full-scan mass analysis from *m*/*z* 100–*m*/*z* 1000 with a resolution of 30,000 in both positive and negative ion modes. The collision energy for collision induced dissociation (CID) was adjusted to 40% of the maximum. The dynamic exclusion (DE) was used to prevent duplication. The repeat count was set at 5 and the dynamic repeat time was 30 s with a dynamic exclusion duration at 60 s. In addition, MSn stages of the obtained datasets were employed by the parent ion list (PIL)-DE dependent acquisition mode.

### 4.5. Data Processing

A Thermo Xcalibur 2.1 workstation was used for data acquisition and data processing. In order to acquire as many fragment ions as possible, we selected the peaks with an intensity over 10,000 for the negative ion mode and 40,000 for the positive ion mode to identify the HTF metabolites. Based on the accurate mass, potential element compositions, and occurrence of possible reactions, the predicted atoms for chemical formulas of all of the deprotonated and protonated molecular ions were set as follows: C [0–30], H [0–50], O [0–20], S [0–2], N [0–3], and ring double bond (RDB) equivalent value [0–15]. The maximum mass errors between the measured and calculated values were fixed within 5 ppm. All relevant data including peak number, retention time, accurate mass, the predicted chemical formula, and corresponding mass error were recorded.

## 5. Conclusions

In the present study, a DMC-based method combined with multiple MS acquisition data processing methods including data-acquisition approaches, data-processing techniques, and structural elucidation approaches provides a powerful integrated methodology where drug metabolites can be predicted, analyzed, and determined completely. The effectiveness of the DMC methodology based analysis was investigated by analyzing HTF for the first time. After evaluating the wholeness of HTF drug metabolites, a total of 106 metabolites (19 potential metabolites included) were identified. Additionally, the proposed in vivo metabolic pathways of HTF consisted of hydrogenation, methylation, demethylation, methoxylation, glucuronide conjugation, sulfate conjugation, ring-cleavage, and their composite metabolic reactions. The present results progressively accomplished a drug metabolism profile from the initial metabolites of the already known data analysis along metabolism clusters, and finally fulfilled the identification and screening of unknown metabolites, which indicates that the proposed methodology is reliable for use to discover and identify drug-related constituents.

## Figures and Tables

**Figure 1 molecules-24-03278-f001:**
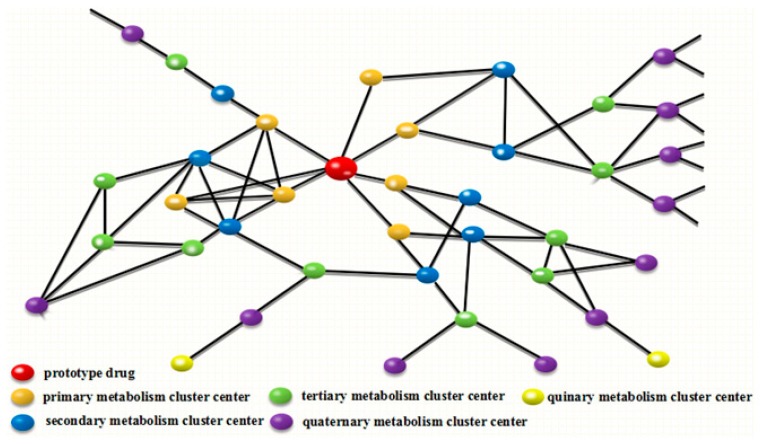
Creation and visualization of drug metabolite clusters (DMCs). The DMCs are constructed from the alignment of metabolite cluster centers to one another. Edges connecting nodes are defined by the MS/MS spectra that determine the relation of two metabolites. With centers stepwise increasing, metabolite clusters have increasingly developed and even intersect among every cluster.

**Figure 2 molecules-24-03278-f002:**
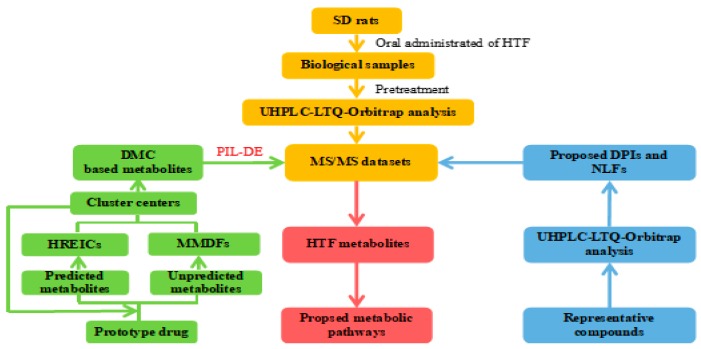
Summary diagram of the developed strategy and methodology.

**Figure 3 molecules-24-03278-f003:**
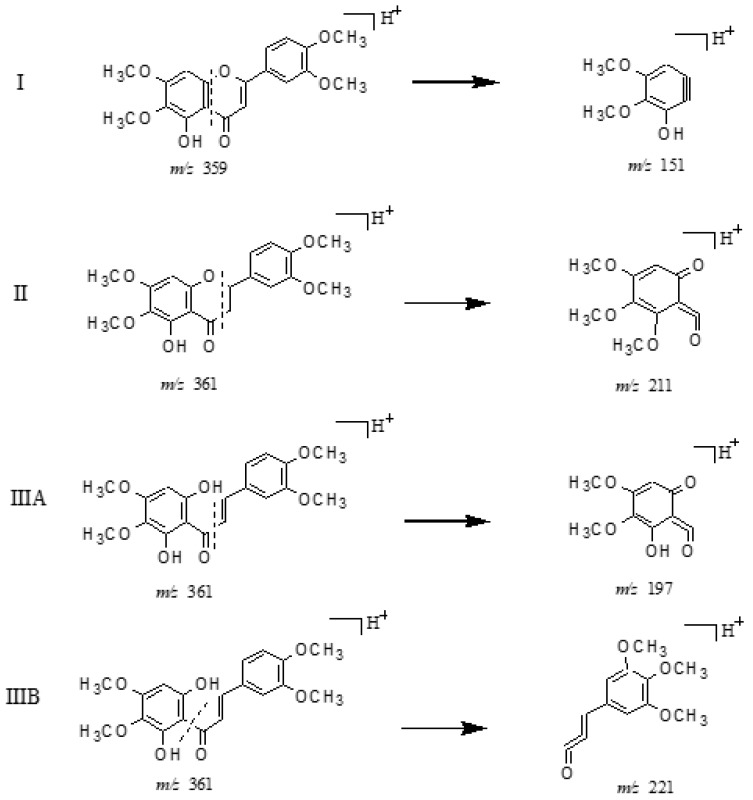
The proposed fragmentation pathways for polymethoxyflavones, polymethoxyflavanone, and polymethoxychalcone derivatives.

**Figure 4 molecules-24-03278-f004:**
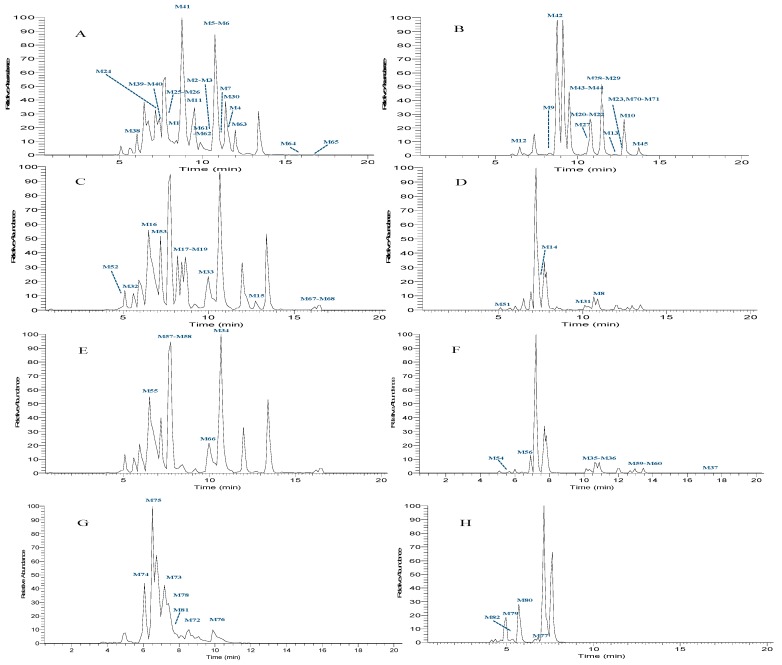
High resolution extracted ion chromatograms for HTF metabolites in rat urine in both negative and positive ion modes. (**A**,**B**) HREIC chromatograms of polymethoxyflavones in negative and positive ion modes; (**C**,**D**) HREIC chromatograms of polymethoxyflavanones in negative and positive ion modes; (**E**,**F**) HREIC chromatograms of polymethoxychalcone in negative and positive ion modes; (**G**,**H**) HREIC chromatograms of conjugate-metabolites in negative and positive ion modes.

**Figure 5 molecules-24-03278-f005:**
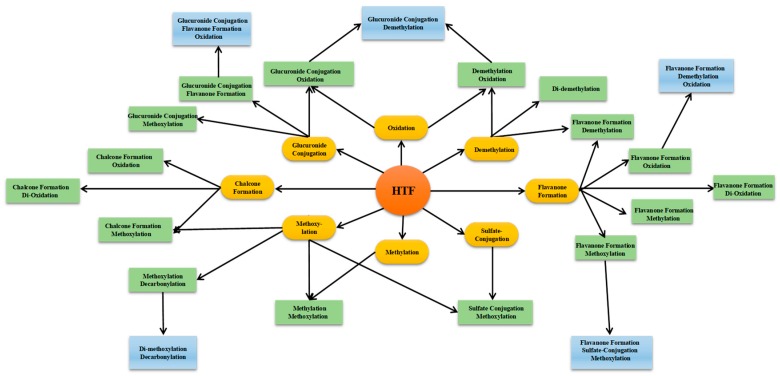
The proposed DMC of HTF in in vivo metabolic patterns in rats.

**Figure 6 molecules-24-03278-f006:**
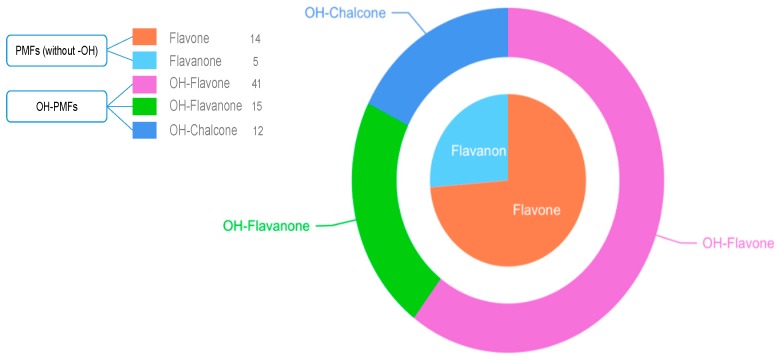
The classification of HTF metabolites.

**Figure 7 molecules-24-03278-f007:**
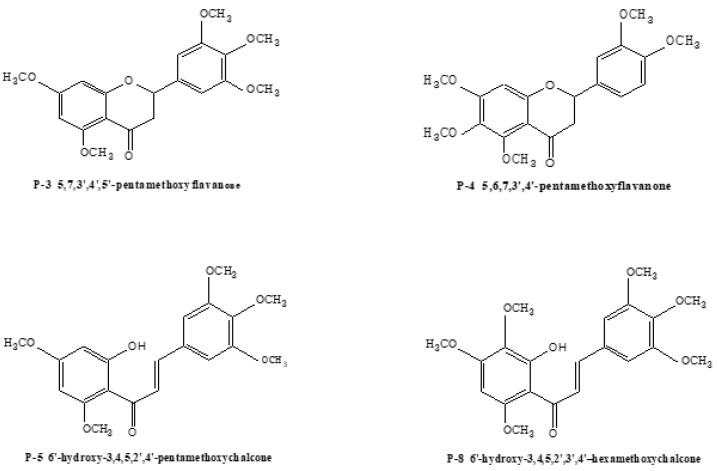
Structures of the nine PMFs reference standards.

**Table 1 molecules-24-03278-t001:** Identification of nine PMFs reference standards in positive mode.

No.	Formula [M + H]^+^	Theoretical Mass *m*/*z*	Experimental Mass *m*/*z*	Error (ppm)	MS/MS Fragment Ions
**P-1**	C_19_H_19_O_8_	359.11256	359.11290	1.0	MS2[359]:326(100),344(71)
					MS3[326]:298(100),278(0.6)
**P-2**	C_18_H_17_O_6_	329.10196	329.10123	−2.2	MS2[329]:314(100),313(66),285(21),268(4),192(0.5)
					MS3[314]:283(100),298(37),167(24),270(24),173(2)
**P-3**	C_20_ H_19_O_7_	375.12817	375.12839	0.5	MS2[375]:211(100),191(18),357(13)
					MS3[211]:196(100),183(29),178(29),150(22)
**P-4**	C_20_H_23_O_7_	375.14382	375.14420	1.0	MS2[375]:211(100),191(43),357(20)
					MS3[211]:196(100),178(24),183(9)
**P-5**	C_20_H_23_O_7_	375.14382	375.14420	1.0	MS2[375]:221(100),181(9)
					MS3[221]:193(100),190(51),191(37),206(26)
**P-6**	C_20_ H_21_O_8_	389.12309	389.12289	−0.3	MS2[389]:359(100),328(70),374(66),356(42),328(15)
					MS3[359]:344(100),341(72),343(66),191(39)
**P-7**	C_20_ H_21_O_8_	389.12309	389.12271	−0.9	MS2[389]:374(100),328(95),359(43),356(14)
					MS3[374]:345(100),312(81),358(29),341(26)
**P-8**	C_21_H_25_O_8_	405.15439	405.15430	−0.2	MS2[405]:221(100),387(26),211(17)
					MS3[221]:193(100),190(63),191(39),206(15)
**P-9**	C_20_ H_21_O_8_	419.13365	419.13360	−0.1	MS2[419]:389(100),404(66)
					MS3[389]:356(100),371(90),361(85),359(79),374(79)

**Table 2 molecules-24-03278-t002:** Identification of HTF metabolites in rat urine and plasma.

Peak	Ion Mode	t_R_ /min	Formula	Theoretical Mass *m*/*z*	Experimental Mass *m*/*z*	Error (ppm)	MS/MS Fragment Ions	Identification/Reactions
**M0**	N	8.86	C_19_H_17_O_7_	357.09692	357.09698	0.2	MS^2^[357]:342(100),327(7),314(2),283(2),284(1),282(0.6),324(0.4)	HTF
							MS^3^[342]:327(100),312(4),299(1),324(1),314(0.6),296(0.5)	
	P	8.86	C_19_H_19_O_7_	359.11256	359.11218	−0.9	MS^2^[359]:298(100),344(71),316(28),329(24),326(22),343(14)	
							MS^3^[298]:283(100),255(18),297(17),270(16),282(14),151(2)	
**M1**	N	7.94	C_17_H_13_O_7_	329.06552	329.06497	−1.8	MS^2^[329]:314(100),299(6),285(1),315(0.6)	Loss of 2CH_2_
							MS^3^[314]:299(100),313(49),285(47),241(4),286(3),296(3)	
**M2**	N	10.01	C_17_H_13_O_7_	329.06552	329.06628	2.1	MS2[329]:229(100),211(76),311(29),293(24),171(21)	Loss of 2CH_2_
							MS3[229]:211(100),209(54),125(21),155(18),167(15),127(12)	
**M3**	N	10.24	C_17_H_13_O_7_	329.06552	329.06714	4.7	MS2[329]:171(100),229(56),293(54),311(49),211(40),	Loss of 2CH_2_
							MS3[171]:127(100),153(74),125(37)	
**M4**	N	11.65	C_17_H_13_O_7_	329.06552	329.06677	3.6	MS2[329]:171(100),201(77),311(56),275(40),293(37),213(7),185(4)	Loss of 2CH_2_
							MS3[171]:127(100),153(71),125(39)	
**M5**	N	10.77	C_18_H_15_O_7_	343.08117	343.08008	−3.3	MS2[343]:328(100),313(3)	Loss of CH_2_
							MS3[328]:313(100),312(25),299(4),300(2),151(1)	
	P	10.77	C_18_H_17_O_7_	345.09687	345.09579	−3.1	MS2[345]:330(100),312(96),284(5),327(1)	
**M6**	N	10.97	C_18_H_15_O_7_	343.08117	343.08066	−3.3	MS2[343]:328(100),329(6),313(5),315(1)	Loss of CH_2_
							MS3[328]:313(100),312(10),299(3),300(2),285(1),298(1)	
	P	10.97	C_18_H_17_O_7_	345.09687	345.09558	−3.7	MS2[345]:312(100),330(93),284(5),327(1)	
**M7**	N	11.26	C_18_H_15_O_7_	343.08117	343.08118	−0.1	MS2[343]:328(100),313(5),315(1)	Loss of CH_2_
							MS3[328]:313(100),312(15),282(14),309(10),299(7),300(2),151(1)	
	P	11.26	C_18_H_17_O_7_	345.09687	345.09546	−4.1	MS2[345]:330(100),312(94),284(18),315(3),329(3),327(1)	
**M8**	P	10.65	C_18_H_19_O_7_	347.11247	347.11169	−2.4	MS2[347]:197(100),177(16),145(4),305(3),182(3),348(2),223(2)	Loss of CH_2_ + Flavanone Formation
							MS3[197]:182(100),164(19),165(5),136(3),137(2)	
**M9**	P	8.37	C_19_H_19_O_7_	359.11256	359.11200	−1.4	MS2[359]:344(100),326(67),343(2),298(2),327(1)	HTF isomer
							MS3[344]:326(100),329(2),298(2),315(1)	
**M10**	P	12.83	C_19_H_19_O_7_	359.11256	359.11142	−3.0	MS2[359]:326(100),344(73),343(8),298(5),327(2)	HTF isomer
							MS3[326]:298(100),299(1)	
**M11**	N	8.97	C_18_H_1__5_O_8_	359.07721	359.0755	−1.7	MS2[359]:344(100),343(9),329(1)	Loss of CH_2_ + Oxidation
							MS3[344]:329(100),328(21),315(4),316(2),314(2)	
**M12**	P	6.40	C_19_H_21_O_7_	361.12817	361.12750	−1.8	MS2[361]:197(100),191(40),343(25),329(9),301(6),182(3)	Loss of Carbonyl + Methoxylation
							MS3[197]:182(100),165(18),123(5),169(5),137(4),151(4)	
**M13**	P	12.57	C_19_H_21_O_7_	361.12817	361.12738	−2.2	MS2[361]:197(100),191(39),211(38),343(9),223(7),177(6),329(4)	Loss of Carbonyl + Methoxylation
							MS3[197]:182(100),165(17),164(9),169(8),151(2)	
**M14**	P	7.33	C_19_H_21_O_7_	361.12817	361.12769	−1.3	MS2[361]:211(100),177(16),329(6),343(6),145(4),196(4)	Flavanone Formation
							MS3[211]:196(100),178(22),150(5),151(3),183(2),179(1)	
**M15**	N	12.23	C_19_H_19_O_7_	359.11362	359.11261	0.2	MS2[359]:149(100),134(35),344(20),343(2),329(2),175(2),179(1)	Flavanone Formation
							MS3[149]:134(100)	
	P	12.25	C_19_H_21_O_7_	361.12817	361.12769	−1.9	MS2[361]:211(100),177(16),343(7),329(6),145(4),196(4)	
							MS3[211]:196(100),178(25),150(5),183(4),151(3)	
**M16**	N	6.40	C_18_H_17_O_8_	361.09182	361.09155	−0.6	MS2[361]:346(100),343(9),181(6),166(4),207(2)	Loss of CH_2_ + Oxidation and Flavanone Formation
							MS3[346]:180(100),166(77),290(54),328(50),331(43),152(26)	
**M17**	N	8.20	C_18_H_17_O_8_	361.09182	361.09155	−0.6	MS2[361]:346(100),343(9),181(6),166(4),207(2),328(1)	Loss of CH_2_ + Oxidation and Flavanone Formation
							MS3[346]:180(100),166(77),290(54),328(50),331(43),318(38)	
**M18**	N	8.42	C_18_H_17_O_8_	361.09182	361.09152	−0.7	MS2[361]:195(100),346(43),180(41),165(37),150(13),343(12),191(5)	Loss of CH_2_ + Oxidation and Flavanone Formation
							MS3[195]:180(100),127(1)	
	P	8.46	C_18_H_19_O_8_	363.10744	363.1066	−2.3	MS2[363]:193(100),197(94),345(51),161(23),133(15),364(7),182(2)	
							MS3[193]:161(100),133(14),178(2),165(1)	
**M19**	N	8.65	C_18_H_17_O_8_	361.09182	361.09137	−1.1	MS2[361]:346(100),195(48),180(19),165(17),362(11),150(6)	Loss of CH_2_ + Oxidation and Flavanone Formation
							MS3[346]:331(100),166(14),328(12),313(4),297(4),303(3)	
**M20**	P	10.64	C_20_H_21_O_7_	373.12822	373.12738	−2.1	MS2[373]:312(100),358(62),329(30),343(27),340(22),357(18),339(3)	Methylation
							MS3[312]:297(100),296(79),151(42),269(36),268(35),281(17),311(15)	
**M21**	P	10.76	C_20_H_21_O_7_	373.12822	373.12726	−2.4	MS2[373]:312(100),358(69),329(28),343(27),340(22),357(18),339(3)	Methylation
							MS3[312]:297(100),268(53),296(46),269(45),151(40),284(17),311(15)	
**M22**	P	10.92	C_20_H_21_O_7_	373.12822	373.12701	−3.1	MS^2^[373]:312(100),358(73),329(28),343(27),340(20),357(15),313(12)	Methylation
**M23**	P	12.71	C_20_H_21_O_7_	373.12822	373.12619	−4.3	MS2[373]:355(100),337(8),319(4)	Methylation
							MS3[355]:337(100),319(60),227(50),213(41),241(32),145(31),173(20)	
**M24**	N	7.42	C_19_H_17_O_8_	373.09172	373.09167	−0.3	MS2[373]:358(100),357(7),343(6)	Oxidation
							MS3[358]:343(100),328(14),330(5),340(1),315(1),329(1)	
**M25**	N	8.02	C_19_H_17_O_8_	373.09172	373.09094	−2.2	MS2[373]:358(100),343(6),357(2),313(1),269(1),295(1)	Oxidation
							MS3[358]:343(100),330(2),327(1)	
**M26**	N	8.61	C_19_H_17_O_8_	373.09172	373.08969	−4.6	MS2[373]:358(100),313(1),357(1)	Oxidation
							MS3[358]:343(100),330(8),329(1)	
**M27**	P	10.44	C_19_H_19_O_8_	375.10737	375.1066	−2.2	MS2[375]:314(100),345(24),360(13),342(12)	Oxidation
							MS3[314]:286(100),285(42),299(5),287(4)	
**M28**	N	10.71	C_19_H_17_O_8_	373.09172	373.09152	−0.7	MS^2^[373]:358(100),98(4),175(4),190(4),124(4)	Oxidation
	P	10.71	C_19_H_19_O_8_	375.10737	375.10645	−2.6	MS^2^[375]:342(100),360(74),375(12),314(10)	
							MS^3^[342]:314(100),315(0.6)	
**M29**	N	11.38	C_19_H_17_O_8_	373.09172	373.09094	−2.2	MS2[373]:358(100),374(26),373(6),359(5),343(3)	Oxidation
							MS3[358]:343(100),329(1),341(0.5)	
	P	11.37	C_19_H_19_O_8_	375.10737	375.10641	−2.7	MS2[375]:360(100),376(38),359(38),342(36),375(29),314(11)	
							MS3[360]:342(100),326(64),344(48),359(46),314(29),327(15),331(10),345(10)	
**M30**	N	11.83	C_19_H_17_O_8_	373.09172	373.09195	0.4	MS2[373]:358(100),343(3),357(1),359(1)	Oxidation
							MS3[358]:343(100),342(7),328(2),299(1),284(1)	
**M31**	P	9.19	C_20_H_23_O_7_	375.14327	375.14301	−2.1	MS2[375]:211(100),191(43),357(23),376(22),343(7)	Methylation + Flavanone Formation
							MS3[211]:196(100),178(24),183(19),151(4),179(2)	
**M32**	N	5.94	C_19_H_19_O_8_	375.10747	375.10773	0.7	MS2[375]:165(100),150(39),360(10),191(3),325(1)	Oxidation + Flavanone Formation
							MS3[165]:150(100),123(0.2),121(0.1)	
	P	6.00	C_19_H_21_O_8_	377.12312	377.12222	−2.3	MS2[377]:359(100),193(73),211(67),161(19),133(14)	
							MS3[359]:300(100),299(95),327(67),328(52),344(36),331(35),313(29)	
**M33**	P	10.15	C_19_H_21_O_8_	377.12312	377.12198	−2.9	MS^2^[377]:207(100),197(75),359(29),175(23),159(17),182(2)	Oxidation + Flavanone Formation
							MS^3^[207]:175(100),145(7),192(2),119(2)	
**M34**	N	10.67	C_19_H_19_O_8_	375.10747	375.10703	−1.1	MS^2^[375]:360(100),345(4),361(4),166(2)	Oxidation + Chalcone Formation
							MS^3^[360]:345(100),166(46),332(14),179(11),304(11),317(10)	
	P	10.67	C_19_H_21_O_8_	377.12312	377.12186	−3.2	MS^2^[377]:221(100),209(38),183(28),343(19),361(17),359(11)	
							MS^3^[221]:193(100),150(58),206(45),191(42),189(19),178(5)	
**M35**	P	10.98	C_19_H_21_O_8_	377.12312	377.12225	−2.2	MS2[377]:197(100),207(82),359(33),175(21),221(14)	Oxidation + Chalcone Formation
							MS3[197]:182(100),164(21),165(5),137(3)	
**M36**	P	11.14	C_19_H_21_O_8_	377.12312	377.12173	−3.6	MS2[377]:197(100),207(84),221(54),359(34),209(19),175(14),183(14)	Oxidation + Chalcone Formation
							MS3[197]:182(100),164(21),165(9),137(3),153(2)	
**M37**	P	17.02	C_19_H_21_O_8_	377.12312	377.12207	−2.7	MS2[377]:359(100),291(99),289(20),221(19),207(17),197(17),175(3)	Oxidation + Chalcone Formation
							MS3[359]:327(100),267(63),341(51),331(45),278(30)	
**M38**	N	6.05	C_20_H_19_O_8_	387.10737	387.10733	−0.2	MS2[387]:372(100),357(4)	Methoxylation
							MS3[372]:357(100),341(0.5),343(0.5)	
	P	6.10	C_20_H_21_O_8_	389.12302	389.12207	−2.6	MS2[389]:328(100),374(66),345(29),359(28),356(22),373(20)	
							MS3[328]:313(100),267(82),312(66),211(64),297(53),284(37)	
**M39**	N	7.30	C_20_H_19_O_8_	387.10737	387.10739	−0.1	MS2[387]:372(100),357(2)	Methoxylation
							MS3[372]:357(100),341(1),343(0.6)	
**M40**	N	7.60	C_20_H_19_O_8_	387.10737	387.10764	0.5	MS2[387]:372(100),357(5),151(1)	Methoxylation
							MS3[372]:357(100),341(1),343(0.6)	
**M41**	N	8.76	C_20_H_19_O_8_	387.10737	387.10693	−1.3	MS2[387]:372(100),388(29),357(9),373(3)	Methoxylation
							MS3[372]:357(100),341(1),343(0.5)	
	P	8.76	C_20_H_21_O_8_	389.12302	389.12198	−2.8	MS2[389]:328(100),374(57),390(47),345(28),359(27),355(3)	
							MS3[328]:313(100),312(78),267(69),211(48),297(41),296(33),300(28)	
**M42**	P	9.08	C_20_H_21_O_8_	389.12302	389.12207	−2.6	MS^2^[389]:374(100),356(75),328(19),390(15)	Methoxylation
							MS^3^[374]:356(100),328(2)	
**M43**	N	9.50	C_20_H_19_O_8_	387.10737	387.10721	−0.6	MS2[387]:372(100),357(2),373(1)	Methoxylation
							MS3[372]:357(100),343(1),341(0.6)	
	P	9.50	C_20_H_21_O_8_	389.12302	389.12207	−2.6	MS2[389]:374(100),328(63),373(29),345(20),356(16),359(15)	
							MS3[374]:328(100),312(44),358(38),345(27),359(17),373(16),356(16)	
**M44**	N	9.80	C_20_H_19_O_8_	387.10737	387.10748	0.1	MS2[387]:372(100),357(3),373(1)	Methoxylation
							MS3[372]:357(100),341(1),343(0.5)	
**M45**	P	13.75	C_20_H_21_O_8_	389.12302	389.1221	−2.5	MS2[389]:356(100),328(71),374(69),390(42),389(14),359(14)	Methoxylation
							MS3[356]:328(100),313(1)	
**M46**	P	15.68	C_20_H_21_O_8_	389.12302	389.12225	−2.1	MS^2^[389]:374(100),356(60),328(57),359(16),375(14),345(13),373(13),147(1)	Methoxylation
**M48**	P	17.76	C_20_H_21_O_8_	389.12302	389.12222	−2.2	MS^2^[389]:374(100),328(52),356(50),359(12),345(12),119(1),211(1)	Methoxylation
**M49**	P	19.47	C_20_H_21_O_8_	389.12302	389.12241	−1.7	MS^2^[389]:374(100),328(56),356(49),359(12),345(12),139(1),158(1)	Methoxylation
**M50**	P	25.00	C_20_H_21_O_8_	389.12302	389.12231	−2.0	MS^2^[389]:374(100),328(51),356(50),373(15),359(14),345(12),185(2)	Methoxylation
**M51**	P	5.19	C_20_H_23_O_8_	391.13867	391.13779	−2.4	MS2[391]:207(100),211(66),373(66),175(22),147(7)	Methoxylation + Flavanone Formation
							MS3[207]:195(100),147(5),119(4),179(3),177(1),192(1)	
**M52**	N	5.61	C_20_H_21_O_8_	389.12412	389.12289	−0.5	MS2[389]:374(100),359(25),341(8),371(5),165(2),307(2)	Methoxylation + Flavanone Formation
							MS3[374]:359(100),179(2),165(2),208(1)	
**M53**	N	7.19	C_20_H_21_O_8_	389.12412	389.12259	−1.2	MS2[389]:179(100),374(73),164(33),373(24),359(14),149(13),121(1)	Methoxylation + Flavanone Formation
							MS3[179]:164(100),149(1)	
**M54**	P	5.72	C_20_H_23_O_8_	391.13867	391.13809	−1.6	MS2[391]:221(100),373(49),207(34),197(18),175(10)	Methoxylation + Chalcone Formation
							MS3[221]:193(100),190(49),206(43),191(41),189(9)	
**M55**	N	6.5	C_20_H_21_O_8_	389.12412	389.12259	−1.2	MS2[389]:374(100),359(36),390(19),165(3),180(3),389(3)	Methoxylation + Chalcone Formation
							MS3[374]:359(100),165(3),179(3),208(1)	
**M56**	N	6.96	C_20_H_21_O_8_	389.12412	389.12292	−0.4	MS2[389]:374(100),359(25),165(3),180(2),373(2),347(1)	Methoxylation + Chalcone Formation
							MS3[374]:359(100),165(3),180(2)	
	P	6.93	C_20_H_23_O_8_	391.13867	391.13785	−2.2	MS2[391]:221(100),373(42),197(28),392(10),193(5),190(5),206(2),191(2)	
							MS3[221]:193(100),190(48),191(41),206(36),189(15)	
**M57**	N	7.64	C_20_H_21_O_8_	389.12412	389.12286	−0.6	MS2[389]:374(100),359(78),390(43),389(19),165(10),375(9),360(7),180(6),208(4)	Methoxylation + Chalcone Formation
							MS3[374]:359(100),165(3),179(3),208(1)	
	P	7.64	C_20_H_23_O_8_	391.13867	391.13779	−2.4	MS2[391]:221(100),373(18),197(17),193(4),190(3),182(2),206(1),191(1)	
							MS3[221]:193(100),190(59),206(47),191(39),189(16),178(5)	
**M58**	N	7.78	C_20_H_21_O_8_	389.12412	389.12296	−0.3	MS2[389]:374(100),359(29),165(3),180(2),347(1)	Methoxylation + Chalcone Formation
							MS3[374]:359(100),165(4),180(2),343(1),208(1),221(1)	
**M59**	P	12.97	C_20_H_23_O_8_	391.13867	391.13809	−1.6	MS2[391]:221(100),197(48),373(12),223(8),349(4)	Methoxylation + Chalcone Formation
							MS3[221]:193(100),190(50),191(40),206(39),189(13),177(6)	
**M60**	N	13.44	C_20_H_21_O_8_	389.12412	389.12305	−0.1	MS2[389]:374(100),359(39),390(32),389(7),208(4)	Methoxylation + Chalcone Formation
							MS3[374]:359(100),208(6),165(3),358(2),180(2),193(1)	
	P	13.44	C_20_H_23_O_8_	391.13867	391.13776	−2.5	MS2[391]:221(100),197(20),373(19),392(19),373(9)	
							MS3[221]:193(100),190(54),205(40),191(40),189(15),178(3)	
**M61**	N	9.87	C_20_H_21_O_8_	389.12412	389.12305	−0.1	MS2[389]:374(100),359(26),208(3),373(2),280(1)	Loss of Carbonyl + Di-Methoxylation
							MS3[374]:359(100),208(4),165(3),180(3),193(1)	
**M62**	N	10.34	C_20_H_21_O_8_	389.12412	389.12292	−0.4	MS2[389]:374(100),359(24),208(2),179(1),165(1)	Loss of Carbonyl + Di-Methoxylation
							MS3[374]:359(100),208(4),165(3),180(2)	
**M63**	N	11.99	C_20_H_21_O_8_	389.12412	389.12314	0.1	MS2[389]:179(100),164(33),374(21),149(14),359(2),121(2),205(2)	Loss of Carbonyl + Di-Methoxylation
							MS3[179]:164(100),149(2)	
**M64**	N	14.97	C_20_H_21_O_8_	389.12412	389.12323	0.3	MS2[389]:343(100),353(16),345(12),313(10),374(10),327(9),179(1),195(1)	Loss of Carbonyl + Di-Methoxylation
							MS3[343]:325(100),299(20),259(17),287(15),271(11),187(10)	
**M65**	N	16.10	C_20_H_21_O_8_	389.12412	389.12418	2.7	MS2[389]:345(100),327(61),343(41),353(27),311(13),151(1)	Loss of Carbonyl + Di-Methoxylation
							MS3[345]:327(100),311(9),325(7),343(7),317(2)	
**M66**	N	10.06	C_19_H_19_O_9_	391.10342	391.1022	−0.4	MS2[391]:155(100),375(78),140(20),360(16),376(15),221(11),169(7)	Di-Oxidation + Chalcone Formation
							MS3[155]:140(100),125(15),123(3),95(3)	
**M67**	N	16.11	C_19_H_19_O_9_	391.10342	391.10046	−4.8	MS2[391]:345(100),347(44),327(37),392(37),329(33),355(31),301(13),343(7)	Di-Oxidation + Flavanone Formation
							MS3[345]:327(100),325(28),311(10),259(9),343(8),341(5)	
**M68**	N	16.32	C_19_H_19_O_9_	391.10342	391.10040	−3.0	MS2[391]:345(100),347(44),327(37),329(33),355(31),343(7)	Di-Oxidation + Flavanone Formation
							MS3[345]:327(100),325(22),311(9),259(5),285(2),301(2)	
**M69**	P	10.54	C_21_H_23_O_8_	403.13877	403.13754	−2.9	MS2[403]:373(100),388(55),342(41),370(8),387(6),289(4)	Methylation + Methoxylation
							MS3[373]:345(100),340(64),343(31),358(25),312(17),181(7)	
**M70**	P	11.50	C_21_H_23_O_8_	403.13877	403.13748	−3.1	MS2[403]:342(100),388(48),343(28),359(24),370(23),343(5),327(3)	Methylation + Methoxylation
							MS3[342]:327(100),281(62),309(29),151(18),312(18),195(9)	
**M71**	P	12.47	C_21_H_23_O_8_	403.13877	403.13745	−3.2	MS2[403]:342(100),388(55),373(29),359(25),387(15),343(6),327(2)	Methylation + Methoxylation
							MS3[342]:327(100),281(78),309(20),314(16),298(16),151(13),195(3)	
**M72**	N	8.54	C_19_H_17_O_11_S	453.04857	453.04858	−0.1	MS2[453]:373(100),358(3)	Sulfate Conjugation
							MS3[373]:358(100),343(0.6)	
**M73**	N	7.19	C_20_H_19_O_11_S	467.06417	467.06430	−0.1	MS2[467]:387(100),372(8),388(8),452(1)	Methoxylation + Sulfate Conjugation
							MS3[387]:372(100),357(1)	
**M74**	N	5.94	C_20_H_21_O_11_S	469.08092	469.08047	1.1	MS2[469]:389(100),259(26),371(6),341(5),374(5),454(4),179(2)	Methoxylation + Sulfate Conjugation and Flavanone Formation
							MS3[389]:341(100),374(96),371(76),340(33),326(18)	
**M75**	N	6.50	C_20_H_21_O_11_S	469.08092	469.07980	−0.23	MS2[469]:389(100),470(16),390(14),374(1)	Methoxylation + Sulfate Conjugation and Flavanone Formation
							MS3[389]:372(100),357(1)	
**M76**	N	9.87	C_20_H_21_O_11_S	469.08092	469.07996	0.1	MS2[469]:389(100)	Methoxylation + Sulfate Conjugation and Flavanone Formation
							MS3[389]:374(100),359(24),208(2),180(1)	
**M77**	N	7.05	C_24_H_23_O_13_	519.11332	519.11322	−0.1	MS2[519]:343(100),175(7),501(5),328(5),329(2)	Loss of CH_2_ + Glucuronide Conjugation
							MS3[343]:328(100),284(0.1),313(0.1)	
	P	7.07	C_24_H_25_O_13_	521.12897	521.12762	−2.5	MS2[521]:345(100),346(14),522(7),330(1)	
							MS3[345]:330(100),312(89),284(3),345(2),327(2)	
**M78**	N	7.49	C_25_H_25_O_13_	533.12897	533.12878	−0.3	MS^2^[533]:357(100),175(2),342(0.5)	Glucuronide Conjugation
							MS^3^[357]:342(100),327(14)	
**M79**	N	4.99	C_25_H_27_O_13_	535.14572	535.14508	0.8	MS^2^[533]:359(100),345(96),175(43),517(22),359(19),212(6),147(6),197(6)	Glucuronide Conjugation + Flavanone Formation
**M80**	N	5.75	C_25_H_25_O_14_	549.12387	549.12384	−0.1	MS2[549]:373(100),358(2),487(0.5),353(0.4)	Oxidation + Glucuronide Conjugation
							MS3[373]:358(100),343(3),286(0.6),297(0.2)	
	P	5.80	C_25_H_27_O_14_	551.13952	551.13818	−2.4	MS2[551]:375(100),191(5),375(3),389(1)	
							MS3[375]:314(100),360(73),331(29),345(26),359(26),342(21)	
**M81**	N	7.53	C_25_H_25_O_14_	549.12387	549.1236	−0.5	MS^2^[549]:373(100),358(2),487(0.5),353(0.4)	Oxidation + Glucuronide Conjugation
							MS^3^[373]:358(100),343(0.4),314(0.1)	
**M82**	N	4.90	C_25_H_27_O_14_	551.14062	551.14038	1.5	MS2[551]:375(100),361(7),367(6),175(1)	Oxidation + Glucuronide Conjugation and Flavanone Formation
							MS3[375]:165(100),150(42),360(11),331(4),316(4),343(3)	
	P	4.89	C_25_H_29_O_14_	553.15517	553.15369	1.8	MS2[553]:377(100),211(2),359(0.5),207(0.5)	
							MS3[377]:359(100),193(77),211(67),181(23),133(16),342(21)	
**M83**	N	6.05	C_26_H_27_O_14_	563.13947	563.13898	−0.9	MS2[563]:387(100),388(24),372(3),175(1)	Methoxylation + Glucuronide Conjugation
							MS3[387]:372(100),357(5)	
**M84**	N	6.52	C_26_H_27_O_14_	563.13947	563.13855	−1.7	MS2[563]:387(100),388(23),372(21),357(4),175(2)	Methoxylation + Glucuronide Conjugation
							MS3[387]:372(100),357(1)	
**M85**	N	6.73	C_26_H_27_O_14_	563.13947	563.13904	−0.8	MS2[563]:387(100),388(26),373(13),372(10),175(2)	Methoxylation + Glucuronide Conjugation
							MS3[387]:372(100),357(12)	
**M86**	N	6.96	C_26_H_27_O_14_	563.13947	563.13892	−1.0	MS2[563]:387(100),373(7),175(3),372(3)	Methoxylation + Glucuronide Conjugation
							MS3[387]:372(100),357(10)	

**Table 3 molecules-24-03278-t003:** Identification of HTF potential metabolites in rat urine and plasma.

No	Peak	t_R_ /min	Formula [M + H]^+^	Theoretical Mass *m*/*z*	Experimental Mass *m*/*z*	Error (ppm)	Identification/ Reactions
**N1**	P	10.78	C_15_H_11_O_6_	287.05497	287.05466	−1.2	Loss of 3CH_2_ + Demethoxylation
**N2**	P	8.46	C_16_H_13_O_6_	301.07062	301.07025	−1.3	Loss of 2CH_2_ + Demethoxylation
**N3**	P	6.98	C_17_H_15_O_6_	315.08627	315.08646	0.4	Loss of CH_2_ + Methoxylation
**N4**	P	8.84	C_17_H_15_O_6_	315.08627	315.08572	−1.8	Loss of CH_2_ + Methoxylation
**N5**	P	12.83	C_17_H_15_O_6_	315.08627	315.08624	−0.2	Loss of CH_2_ + Methoxylation
**N6**	P	8.78	C_18_H_19_O_6_	331.11761	331.11691	−2.1	Loss of OCH_2_ + Flavanone/Chalcone Formation
**N7**	P	7.96	C_17_H_17_O_7_	333.09697	333.09683	−0.1	Loss of 2CH_2_ + Flavanone/Chalcone Formation
**N8**	P	7.38	C_17_H_13_O_8_	345.06042	345.06033	−0.4	Loss of CH_4_ and CH_2_ + Oxidation
**N9**	P	10.03	C_19_H_21_O_9_	393.11797	393.11673	−3.2	Di-Oxidation + Flavanone/Chalcone Formation
**N10**	N	7.86	C_17_H_13_O_10_S	409.02237	409.02164	−1.8	Loss of 2CH_2_ + Sulfate Conjugation
**N11**	P	9.38	C_17_H_15_O_10_S	411.03802	411.03696	−2.6	Loss of 2CH_2_ +Sulfate Conjugation
**N12**	N	8.19	C_18_H_15_O_10_S	423.03802	423.03772	−0.7	Loss of CH_2_ + Sulfate Conjugation
**N13**	N	4.77	C_18_H_17_O_10_S	425.05472	425.05362	−0.1	Loss of CH_2_ + Flavanone/Chalcone Formation and Sulfate Conjugation
**N14**	P	5.36	C_19_H_21_O_10_S	441.08497	441.08429	−1.5	Flavanone/Chalcone Formation + Sulfate Conjugation
**N15**	N	6.84	C_23_H_21_O_13_	505.09767	505.09750	−0.3	Loss of 2CH_2_ + Glucuronide Conjugation
**N16**	P	6.81	C_23_H_23_O_13_	507.11332	507.11282	−0.9	Loss of 2CH_2_ + Glucuronide Conjugation
**N17**	P	7.21	C_24_H_27_O_12_	507.14971	507.14813	−3.1	Loss of OCH_2_ + Flavanone/Chalcone Formation and Glucuronide Conjugation
**N18**	P	5.57	C_24_H_23_O_13_	519.11332	519.11176	−3.0	Loss of CH_4_ + Glucuronide Conjugation
**N19**	P	6.74	C_24_H_25_O_14_	537.12387	537.12378	−0.1	Loss of CH_2_ + Oxidation and Glucuronide Conjugation

**Table 4 molecules-24-03278-t004:** The positions of the substituent groups of PMF reference standards.

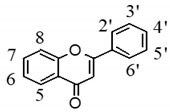					
No	PMFs	Formula	–OH	–OCH_3_	Mass Weight
**P-1**	5-hydroxy-6,7,3′,4′-tetramethoxyflavone	C_19_H_18_O_7_	5	6,7,3′,4′	358
**P-2**	5-hydroxy-7,3′,4′-trimethoxyflavone	C_18_H_16_O_6_	5	7,3′,4′	328
**P-6**	5-hydroxy-6,7,8,3′,4′-pentamethoxyflavone	C_20_H_20_O_8_	5	6,7,8,3′,4′	388
**P-7**	5-hydroxy-6,7,3′,4′,5′-pentamethoxyflavone	C_20_H_20_O_8_	5	6,7,3′,4′,5′	388
**P-9**	5-hydroxy-6,7,8,3′,4′,5′-hexamethoxyflavone	C_21_H_22_O_9_	5	6,7,8,3′,4′,5′	418

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
