# Peer review of "Drug Metabolite Cluster-Based Data-Mining Method for Comprehensive Metabolism Study of 5-hydroxy-6,7,3′,4′-tetramethoxyflavone in Rats"

_molecules, 2019, doi:10.3390/molecules24183278_

Round 1

Reviewer 1 Report

This manuscript entitled " Drug metabolite clusters-based data-mining method for comprehensive metabolism study of 5-hydroxy-6,7,3',4'-tetramethoxyflavone in rats " by Wang et al. described the development of a method called drug metabolite clusters (DMC) by using 5-hydroxy-6,7,3',4'-tetramethoxyflavone as example.

I  recommend this manuscript be accepted for publication. However, before this paper is published some revisions need to be done:

The authors didn’t provide enough information for the cluster construction (Fig 6). The authors just used one compound (HTF) as example to establish this method. More compounds are needed to validate this method. All metabolites identified should be assigned to each cluster center. Figure 4, the fragmentation pathway II was wrong, please check it again

Author Response

Thanks for the reviewers’ and editor’ comments concerning our manuscript entitled Drug metabolite clusters-based data-mining method for comprehensive metabolism study of 5-hydroxy-6,7,3',4'-tetramethoxyflavone in rats (Manuscript ID: molecules-572902 ). Those comments are valuable and helpful for revising and improving our manuscript. We have studied the comments carefully and revised our manuscript. The main corrections in the paper and the responds to the comments were listed in attachment.

If the manuscript has any more questions, please contact us as soon as possible. We will revise it according to your request in time. Thanks for the suggestions you kindly offered again.

Sincerely yours

Yuqi, WANG

School of Chinese Pharmacy, Beijing University of Chinese Medicine, Beijing 102488, China.

E–mail address: wyq0831@bucm.edu.cn

Reviewer 2 Report

Dear authors,

Identification of drug metabolites in a complex biological matric is a bottle next in drug discovery. There are many software available in market (ACD- Intellixtact and MetaSensce) and still many have been proposed as none offer complete solution.

The DMC approach described by the team in this work "Drug metabolite clusters-based data-mining 2 method for comprehensive metabolism study of 3 5-hydroxy-6,7,3',4'-tetramethoxyflavone in rats" is step in right direction especially when dealing with the plant based metabolites. Often these products are not well characterised due to difficulty in obtaining the standards. You have done a wonderful job in picking up the numerous metabolites and demonstrated the utility of DMC. I found you approach is good and applicable. Perhaps you should have validated this approach in rat liver microsome assay and then applied to in-vivo studies to give confidence in this approach in picking up the correct metabolites.

This is an extensive piece of work with so many tables and figures to interpret, makes it harder for reader to under standard. 

Best wishes,

Author Response

Thanks for the reviewers’ and editor’ comments concerning our manuscript entitled Drug metabolite clusters-based data-mining method for comprehensive metabolism study of 5-hydroxy-6,7,3',4'-tetramethoxyflavone in rats (Manuscript ID: molecules-572902 ). Those comments are valuable and helpful for revising and improving our manuscript. We have studied the comments carefully and revised our manuscript. The main corrections in the paper and the responds to the comments were listed as follows:

-Reviewer

The DMC approach described by the team in this work "Drug metabolite clusters-based data-mining 2 method for comprehensive metabolism study of 3 5-hydroxy-6,7,3',4'-tetramethoxyflavone in rats" is step in right direction especially when dealing with the plant based metabolites. Often these products are not well characterised due to difficulty in obtaining the standards. You have done a wonderful job in picking up the numerous metabolites and demonstrated the utility of DMC. I found you approach is good and applicable. Perhaps you should have validated this approach in rat liver microsome assay and then applied to in-vivo studies to give confidence in this approach in picking up the correct metabolites.

Point 1: Perhaps you should have validated this approach in rat liver microsome assay and then applied to in-vivo studies to give confidence in this approach in picking up the correct metabolites. This is an extensive piece of work with so many tables and figures to interpret, makes it harder for reader to under standard.

Response: Thanks for the suggestions you kindly offered. We will take advantage of rat liver microsome assay for validating this approach for the further study. And we have revised part of tables to supplementary materials.

If the manuscript has any more questions, please contact us as soon as possible. We will revise it according to your request in time. Thanks for the suggestions you kindly offered again.

Sincerely yours

Yuqi, WANG

School of Chinese Pharmacy, Beijing University of Chinese Medicine, Beijing 102488, China. 

E–mail address: wyq0831@bucm.edu.cn

Reviewer 3 Report

Dear Editor,

It is my pleasure to review the manuscript, “Drug metabolite clusters-based data-mining

3 method for comprehensive metabolism study of 4 5-hydroxy-6,7,3',4'-tetramethoxyflavone in rats" for molecules. The subject of the paper is interesting. The present work will provide some idea in the field of data mining for different drug molecules. However, I have only concerns that need to be addressed in the manuscript. Is this method is robust for all kind of bioactive compounds or for neurotransmitter and steroids?

Author Response

Thanks for the reviewers’ and editor’ comments concerning our manuscript entitled Drug metabolite clusters-based data-mining method for comprehensive metabolism study of 5-hydroxy-6,7,3',4'-tetramethoxyflavone in rats (Manuscript ID: molecules-572902 ). Those comments are valuable and helpful for revising and improving our manuscript. We have studied the comments carefully and revised our manuscript. The main corrections in the paper and the responds to the comments were listed as follows:

-Reviewer

It is my pleasure to review the manuscript, “Drug metabolite clusters-based data-mining method for comprehensive metabolism study of 4 5-hydroxy-6,7,3',4'-tetramethoxyflavone in rats" for molecules. The subject of the paper is interesting. The present work will provide some idea in the field of data mining for different drug molecules. However, I have only concerns that need to be addressed in the manuscript.

Point 1: Is this method is robust for all kind of bioactive compounds or for neurotransmitter and steroids?

Response: Thanks for the suggestions you kindly offered. Our study is shown that the method available for bioactive compounds, such as flavone, anthraquinone and saponin. However, it is not clear whether the method apply to neurotransmitter and we will remain focused on relevant study.

If the manuscript has any more questions, please contact us as soon as possible. We will revise it according to your request in time. Thanks for the suggestions you kindly offered again.

Sincerely yours

Yuqi, WANG

School of Chinese Pharmacy, Beijing University of Chinese Medicine, Beijing 102488, China. 

E–mail address: wyq0831@bucm.edu.cn
